# Neuroimmune Regulation of Surgery-Associated Metastases

**DOI:** 10.3390/cells10020454

**Published:** 2021-02-20

**Authors:** Michael R. Shurin, James H. Baraldi, Galina V. Shurin

**Affiliations:** 1Department of Pathology, University of Pittsburgh, Pittsburgh, PA 15213, USA; shuringv@upmc.edu; 2Department of Immunology, University of Pittsburgh Medical Center, Pittsburgh, PA 15213, USA; 3Department of Neuroscience, University of Pittsburgh, Pittsburgh, PA 15260, USA; james.baraldi@pitt.edu

**Keywords:** neoneurogenesis, neuroimmune axis, metastasis, neuroglia

## Abstract

Surgery remains an essential therapeutic approach for most solid malignancies. Although for more than a century accumulating clinical and experimental data have indicated that surgical procedures themselves may promote the appearance and progression of recurrent and metastatic lesions, only in recent years has renewed interest been taken in the mechanism by which metastasizing of cancer occurs following operative procedures. It is well proven now that surgery constitutes a risk factor for the promotion of pre-existing, possibly dormant micrometastases and the acceleration of new metastases through several mechanisms, including the release of neuroendocrine and stress hormones and wound healing pathway-associated immunosuppression, neovascularization, and tissue remodeling. These postoperative consequences synergistically facilitate the establishment of new metastases and the development of pre-existing micrometastases. While only in recent years the role of the peripheral nervous system has been recognized as another contributor to cancer development and metastasis, little is known about the contribution of tumor-associated neuronal and neuroglial elements in the metastatic disease related to surgical trauma and wound healing. Specifically, although numerous clinical and experimental data suggest that biopsy- and surgery-induced wound healing can promote survival and metastatic spread of residual and dormant malignant cells, the involvement of the tumor-associated neuroglial cells in the formation of metastases following tissue injury has not been well understood. Understanding the clinical significance and underlying mechanisms of neuroimmune regulation of surgery-associated metastasis will not only advance the field of neuro–immuno–oncology and contribute to basic science and translational oncology research but will also produce a strong foundation for developing novel mechanism-based therapeutic approaches that may protect patients against the oncologically adverse effects of primary tumor biopsy and excision.

## 1. Introduction

About 1.9 million new cancer cases are projected to occur in the USA and almost 20 million cases worldwide in 2021 [1,2]. It was estimated earlier that almost fifty percent of malignant tumors in humans might metastasize prior to clinical presentation [3,4], although the number may be lower and is certainly cancer dependent. While the 5-year survival rate for all cancers combined has increased substantially since the early 1960s, from ~40% to ~70%, outcomes for patients with metastatic cancer have largely remained stagnant [1,5]. Approximately 70–90% of all cancer-related deaths are due to metastases [6,7]. Importantly, observations of accelerated metastatic tumor growth after excision of the primary tumor have been repeatedly reported for several cancers, including colorectal, ovarian, NSCLC, breast, pancreatic, and other cancers (Table 1) [8,9,10,11,12,13,14,15,16,17]. Even the incisional biopsies, including the diagnostic core needle biopsy, might increase local tumor recurrence, lymph node metastases, and distant metastases [18,19,20,21]. However, our understanding of the cellular and molecular pathways of this phenomenon is still limited. Although accumulating evidence unequivocally demonstrates that an intratumoral cellular network actively supports metastatic dissemination of malignant cells [22], clinical data suggest that not all stromal elements and factors regulating migratory and invasive potential of malignant cells have been elucidated. This markedly limits the development and comparative testing of innovative antimetastatic approaches. It is increasingly critical to understand the principles of surgery-induced tumor progression and dissemination to conceive perioperative or adjuvant strategies to further improve long-term tumor control [23]. Understanding the mechanisms by which cancer progression is accelerated as a result of surgery may also provide pharmacologic interventions [24].

Tumor-associated immune cells, fibroblasts, epithelial cells, pericytes, and adipocytes can enhance metastasis by regulating extracellular matrix remodeling and neovascularization, and by promoting the epithelial-mesenchymal transition of the malignant cells [22,59]. Recently, intratumoral neurofilaments have been recognized as important constituents of the tumor milieu [60,61,62], and the degree of tumor innervation has been correlated with metastases and patient survival [63,64,65], Although the involvement of neurotransmitters and neuropeptides in nerve-mediated metastases has been proposed, the role of the neuroglia of the peripheral nervous system (PNS) in promoting metastases of solid tumors remains unconsidered. With the exception of perineural invasion (i.e., locoregional invasion of cancer into the space surrounding a nerve), the extent to which the Schwann cells, the principal glia of the PNS, participate in the formation of metastases has not been elucidated. A recent surge in studies of Schwann cell biology has revealed their expansive functions in neurodegenerative diseases, pain syndrome, autoimmune and inflammatory neuropathies, nerve and tissue repair, tissue regeneration, and cell-based therapy for spinal cord injury and autoimmune neurological diseases. However, the pro-metastatic activity of Schwann cells during surgery/biopsy-associated wound healing has not yet been considered.

New data demonstrates that nerves/Schwann cells are present in human and animal tumor specimens, and that nerves/Schwann cells may accelerate tumor growth and progression in mouse tumor models [66,67,68,69,70,71]. It is conceivable to suggest that this effect is due to cancer cell-Schwann cell-nerve cell crosstalk, which induces the so-called “repair-like” phenotype of Schwann cells and results in: (i) attraction and activation of immune regulatory cells, (ii) reorganization of the extracellular matrix, and (iii) enhanced migratory and invasive potential of malignant cells. New in vivo data suggest that these consequences of tumor-nerve-Schwann cell interactions might increase cancer’s metastatic potential. We can speculate that neuroglial failure occurs when adaptive strategies developed by tumor-activated Schwann cells fail, and, in some settings, when the program associated with Schwann cell-driven tissue repair culminates in a maladaptive response that contributes to metastatic disease. It is possible that tumor-associated “repair-like” Schwann cells promote metastasis during surgery/biopsy-associated wound healing even more strongly since surgical stress and wound healing pathways contribute to the higher level of Schwann cell activation, dedifferentiation, and proliferation in the regenerating tumor microenvironment. However, this has never been experimentally tested.

## 2. The Neuronal Regulation of Tumor Progression

The crosstalk between malignant cells and the stromal and infiltrating cells is fundamental in the tumorigenesis process. Nerve endings are also detected within solid tumors [72,73]. Their communication with tumor cells is believed to represent so-called “neuro-neoplastic synapses” or “tumor-nervous connections” [74,75]. The role of the nerve filaments seen within the tumor mass was first believed to be mechanical, providing “paths” for the migration of the perineural invading cells [76,77]. Perineural invasion, also called perineural spread or neurotropic carcinomatous spread, is malignant cell invasion in, around, and through nerves, which is histologically observed as cancer cells within the layers of the nerve sheath including epineurium, perineurium, and endoneurium. However, it is now clear that the PNS, as a functionally pertinent association of cells and factors at the tumor milieu, regulates tumor development, growth, and dissemination [75,78,79]. Neurons within and at the tumor periphery release neurotransmitters, neuropeptides, and other biologically active substances acting on specific receptors on cancerous cells, stromal elements, and infiltrating immune cells, and altering cell function and cellular interactions in the tumor milieu.

A growing body of evidence shows that cancerous cells can utilize the benefit of the factors released by the nerves to generate a positive environment for survival, proliferation, and spreading [64,80,81,82]. Neural-related factors can alter the progression of metastasis, affecting the base membranes’ degradation and cancerous cell invasiveness, motility, extravasation, and colonization. They also modulate angiogenesis, the tumor stroma, immune cell functions, bone marrow activity, and local and systemic inflammatory pathways to impact metastases [74,83]. These neurotransmitters, neurotrophins, and neuropeptides include acetylcholine, catecholamines, γ-aminobutyric acid, serotonin, substance P, neurokinin A, bombesin, neuropeptide Y, vasoactive intestinal polypeptide, opioids, neurotensin, and other neuromodulating molecules. For instance, activation of β-adrenoceptors on tumor cells and tumor-associated macrophages can promote metastasis in animal models of breast, pancreatic, colon, neuroblastoma, ovarian, and prostate cancers [64,65,84,85,86,87]. Interestingly, in the head and neck cancer model, the crosstalk between malignant cells and neurons represents a mechanism by which tumor-associated neurons can be reprogrammed towards an adrenergic phenotype that augments tumor progression [88]. Although impediment of signaling pathways of the sympathetic nervous system with β-blockers or genetic deletion of β-adrenergic receptors primarily terminates metastasis of different types of tumors, α-adrenergic receptors’ function in cancer metastasis is yet to be clarified [89]. Acetylcholine secreted by tumor-infiltrating nerves from the parasympathetic nervous system was reported to target stromal cells expressing muscarinic receptors and to play a role in modulating tumor cell invasion and migration [64]. Neuropeptide Y, which can be released from sympathetic neurons, is generally accepted to be a powerful angiogenic factor [90]. Substance P and its receptors, at least in breast cancer, have been implicated in bone marrow metastasis formation [91]. Methionine-enkephalin expression in colorectal carcinomas may be associated with nodal and liver metastasis [92], while expression of bombesin/gastrin-releasing peptide in prostate cancer may be related to the lymph node metastases [93].

Although a broad understanding of the complex and multifaceted tumor-regulating role of neuronal factors has been reached [75,94], little is known about the role of tumor innervation in metastasis development in response to surgery and therapy. This gap in our knowledge is important because understanding the role of the PNS in cancer progression and metastasis formation after surgical procedures (tumor biopsy or resection) has a high significance in the clinical management of patients with cancer. In spite of exciting evidence that the removal of nerves from the tumor microenvironment is sufficient to terminate or decelerate disease progression, chemical or surgical denervation is unlikely to be of clinical use [79]. On the other hand, novel clinical studies emphasize cancer patient vulnerability to disease recurrence and metastasis formation following diagnostic and surgical interventions: for instance, excisional or incisional biopsy. The potential magnitude of perioperative vulnerability is underscored by the fact that greater than 60% of nearly 18–20 million patients diagnosed with cancer each year worldwide will require surgical resection [95]. As such, any opportunity to abrogate the risk of cancer progression arising during or after the vulnerable perioperative period could provide substantial benefit to patients. Thus, understanding the role of the PNS in surgery-induced metastasis should support the development of novel clinical approaches to reduce or limit metastasis formation.

## 3. Tissue Damage and Wounding Affect Tumor Progression

For more than a century accumulating clinical and experimental data have indicated that surgical procedures themselves may promote the appearance and progression of malignant lesions (Table 1) [25,29,30,36,40,42,96]. For example, investigations bearing on the occurrence of metastases in mice from which implanted tumors had been removed by operation were made by Clunet at the beginning of the last century [97,98]. Excision of the primary tumor mass might involve obvious dangers because it can release malignant cells into the systemic circulation or lymphatics [99,100,101], up-regulate expression of angiogenic and growth factors [26,102,103,104,105], and inhibit cell-mediated immunity [106,107,108]. The cytokine cascade activated in response to surgical trauma consists of a complex biochemical network with diverse effects, which are essential for wound healing, on the injured host [109,110]. Together, these postoperative outcomes have been suggested to synergistically accelerate the establishment of new distant metastases and the advancement of pre-existing micrometastases [111].

Biopsy remains the gold standard for the diagnosis of many cancers. However, this approach does not address the presence of microscopic residual or “in-transit” malignant cells, which may give rise to metastatic foci [112,113,114,115]. For instance, in patients with residual melanoma after the initial biopsy, 5-year survival is significantly lower, and recurrence is documented in ~25% of patients, with distant metastases being the most common form (40%) [20,53]. Importantly, observations of accelerated metastatic tumor growth after excision of the primary tumor have been repeatedly reported for colorectal, ovarian, lung, breast, gastric, bladder, testicular, and pancreatic cancers (Table 1) [8,9,10,11,12,13,14,15,16,17,31,40,50,54,116,117,118]. Even when complete locoregional control was thought to have been achieved, postoperative disease recurrence occurred in up to one-third of patients [119].

The local tissue response to surgery and biopsy involves the initiation of wound healing, which includes inflammation, neuronal regeneration, extracellular matrix reorganization, and neoangiogenesis (Figure 1). At the same time, these processes are known to be associated with the promotion of tumor growth and metastasis [16,30,36,42,109,120]. For instance, in a zebrafish larval model of Ras(G12V)-driven neoplasia, wound-associated neutrophils increase the proliferation of pre-neoplastic cells, and chronic wounding leads to a higher incidence of melanoma [121]. Furthermore, tumor growth is enhanced in healing wounds but not in the surrounding normal tissues: the probability of a tumor cell leading to a deposit in a wound is increased 1000-fold compared to normal tissue [42,103]. The degree of surgical tissue wounding correlates with the metastatic disease burden in mouse models of colon and breast cancers [48,122], and surgery-induced immune regulatory cells, such as MDSC and Treg cells, accelerate the formation of pulmonary metastases [123]. Macrophages have also been shown to contribute both to post-surgical tumor relapse and growth of metastases, probably by stimulating a population of tumor-initiating cells [52]. Postoperative NK-cell suppression has been reported to correlate with augmented metastatic burden in animal models; in cancer patients, reduced NK-cell activity during the postoperative period has been associated with a high rate of disease recurrence and mortality [49,124,125]. Furthermore, surgery-induced inflammation may potentiate colon cancer stem cell involvement in the metastatic process [126].

These clinical and experimental data suggest that biopsy or surgery-induced wound healing can accelerate the growth of pre-existing micrometastases (i.e., escape from dormancy) or promote survival and invasiveness of residual cancerous cells leading to metastasis [16,127]. Determining mechanisms of cancer spread during wound healing and understanding how peri- and postoperative care should be adapted to reduce the risk of local and distal disease recurrence is an important clinical issue [127,128]. This knowledge is pivotal for the development of therapeutic strategies to prevent or reduce surgery-associated metastasis.

Recently, peripheral nerves and neuroglia have also been implicated in wound healing and cancer spreading [129,130]. For instance, tumor progression after surgical procedures has, in part, been linked to high levels of β-adrenoreceptor signaling [10,48,131]. However, the mechanisms of nerve/glia involvement in an accelerated metastatic disease after excision of the primary tumor have never been explored.

## 4. PNS Functioning in Surgery-Associated Metastasis Formation

Surgical resection is still the most valuable procedure to eliminate primary tumors and involved lymph nodes. However, residual cancerous cells may remain after surgery, and the tumor dislodged during excision may spread via lymphovascular vessels [132,133]. For instance, identification of tumor cells in the blood and peritoneal lavage fluid after surgery has been linked with significantly shorter disease-free survival in patients with colorectal cancer [132]. Similarly, circulating tumor cell numbers expand following surgery for gastric, lung, breast, and hepatocellular cancers and are associated with poor survival [55,57,58,134]. In addition, postoperative wound healing may lead to locoregional and distal disease recurrence by accelerating the growth of pre-existing micrometastases (i.e., escape from dormancy) or by enabling residual tumor cells to spread faster (Figure 1) [16,127,135]. Furthermore, surgery-induced or anesthesia-induced activation of the hypothalamic-pituitary-adrenal (HPA) axis and sympathetic nervous system may facilitate metastasis through the release of neuroendocrine mediators such as catecholamines and prostaglandins, which, in turn, increase immunosuppressive cytokines (e.g., IL-4, IL-10, TGF-β) and VEGF, as well as proinflammatory cytokines (e.g., IL-6 and IL-8), which promote tumor angiogenesis and spreading or creation of the pre-metastatic niches [23,136,137,138,139]. While necessary for normal tissue repair, the local and systemic changes of the innate and adaptive immune responses are able to inspire cancerous cell proliferation, migration, and survival, as well as intratumoral angiogenesis and extravasation of disseminating malignant cells [140]. In addition to the immune cells in the tumor milieu, neurons are now known to be involved in the complex interactions between the formation of cancers, inflammation, wound healing, and metastasis formation.

Surgical tumor removal is commonly associated with a traumatic injury of neuronal axons within the resected tumor or within the surrounding or peripheral tumor tissue. Following peripheral nerve injury, a cascade of molecular and cellular events is initiated at the proximal and distal ends of the lesion site. The distal end is disconnected from the neural body and undergoes Wallerian degeneration [141,142]. Schwann cells (or neurolemmocytes), the principal glial cells of the PNS [143], play a crucial role in the repair of peripheral nerves and represent the major players providing a specialized local extracellular microenvironment for regrowth of injured axons and rebuilding of myelin sheaths on regenerating axons. Distal Schwann cells denervate, i.e., detach from the axon, break down their myelin sheaths, and attract and polarize macrophages to remove axonal and myelin debris [144]. This is associated with Schwann cell de-differentiation/transdifferentiation, proliferation, and production of neurotrophic factors (“repair” Schwann cells) that support and guide the re-growing axon [145,146].

In addition to neurotrauma, new findings suggest that injury can activate a repair program in adult Schwann cells [147], which promotes the repair and regeneration of different tissues [148,149]. Recognition of versatile Schwann cell functions allows us to speculate that tissue wounding, induced by tumor excision or biopsy, may augment the ‘repair’ phenotype of Schwann cells, which is associated with the promotion of malignant cell motility, invasiveness, and metastasis (Figure 1). Interestingly, it was recently discovered that the local neurodegenerative response to cutaneous melanoma growth is quite similar to the neuronal repair pathway induced by skin injury [70,150]. Specifically, genetic, molecular, phenotypic, and functional similarities between tumor-activated, or “repair-like” Schwann cells at the tumor site and nerve injury-induced “repair” Schwann cells were revealed. Furthermore, a new unpublished work has revealed that ex vivo generated tumor-activated and resident PNS-injury-induced (“repair”) Schwann cells accelerated tumor growth and formation of metastases in vivo, while the absence of Schwann cells significantly decelerated tumor growth and metastasis (Y.Bunimovich, UPMC, personal communication). These results, as well as an almost ubiquitous presence of Schwann cells throughout the body, raise an important question: How are Schwann cells involved in the adjustment of the tumor environment after surgery/biopsy? Which mechanisms of tumor-Schwann-immune cell interactions that promote metastasis during wound healing may be involved in this phenomenon? Both in vitro and in vivo studies are needed to answer these questions and uncover the role of neuronal and neuroglial cells in the regulation of metastasis formation after surgical removal of primary tumor tissue.

Interestingly, a novel, recently reported findings demonstrates that Schwann cells can accelerate metastasis in several cancer models [67,68,69]. Although the main function of Schwann cells is to maintain axonal integrity, Schwann cells have been shown to stimulate pancreatic and prostate cancer cell invasion in an integrin-dependent manner and to promote perineural invasion via neural cell adhesion molecule 1 (NCAM1) signaling [151,152]. Moreover, tumor-activated Schwann cells have been shown to activate the CXCL5/CXCR2/PI3K/AKT/GSK-3β/Snail-Twist pathway to promote the epithelial-to-mesenchymal transition, motility, invasiveness, and metastatic potential of lung cancer cells both in vitro and in vivo in immunocompetent mice [68]. In addition to this direct effect of Schwann cells on tumor cells, new data have revealed that in the tumor microenvironment, Schwann cells display a strong ability to chemoattract immature myeloid cells and conventional dendritic cells (DC) and to polarize them into MDSC and regulatory DC that express robust immunosuppressive properties. [69,150] Our new data demonstrate that in addition to affecting myeloid regulatory cells, tumor-activated Schwann cells can also chemoattract T cells and up-regulate their exhaustion phenotype (Shurin et al., unpublished data). Thus, results showing that tumor-activated Schwann cells participate in the development of the immunosuppressive tumor microenvironment, together with the data revealing phenotypic and functional similarities between tumor-activated, i.e., “repair-like,’ Schwann cells and neurotrauma-induced, i.e., “repair,” Schwann cells [70], allow the following speculation. Surgical procedures associated with cancer treatment may provide additional polarization of Schwann cells in the cancer milieu that directly and indirectly (via the immune system) support tumor spreading and dissemination (Figure 1). In other words, the local and systemic crosstalk between the PNS and immune elements and factors is an important regulator not only of cancer development but also of cancer progression and metastasis.

However, the involvement of Schwann cells in metastasis formation as a consequence of wound healing has never been suggested. More experimental results are needed as they can provide a new target for safe and accelerated wound healing and tissue regeneration after oncology-associated surgical procedures.

## 5. Future Directions

Only in recent years has the role of the PNS been recognized as a new contributor to cancer development and metastasis. At the same time, old data showing that the removal of human and experimental animal tumors may be followed by an abrupt increase in metastatic growth (Table 1) [31,50,118,153,154,155,156] have attracted new attention. As 45–60% of patients diagnosed with common cancer require surgery to remove their tumor, as part of their primary cancer treatment [95,157], new questions should now be answered: How may PNS elements be involved in surgery-induced metastasis spreading, and how can they be targeted for improving patients’ survival and wellbeing? Answering these questions will allow us to design innovative approaches to alleviate undesired impacts of surgical procedures and, at the same time, to expand the clinical benefit of these interventions.

Another important point is that in spite of growing data showing some undesired effects of surgical intervention on tumor progression and spread, this does not imply that surgical resection or biopsy should be restricted since these side effects may only affect a subset of patients. The benefit of these surgical procedures for correct diagnosis and expanded survival is indisputable and outweighs the potential negative side effects, justifying their continued use versus their cessation [158]. It remains crucial to completely recognize their mechanistic impact on tumor neuroimmunopathology with the aim of averting these effects and of developing adjuvant remedies that can make these surgical interventions much safer and more beneficial for the patients [158].

The perioperative period is now recognized as crucial in influencing the incidence of postoperative metastases and long-standing cancer outcomes [131,159]. Therefore, numerous perioperative preventive interventions are currently being evaluated. Importantly, new studies provide hope that tumor resection and the wound-healing response can be uncoupled, resulting in attenuation of the outgrowth of distant cancerous cells, especially if this outgrowth is controlled by the immune responses. Although the impact of this approach on prolonging the clinical outcome after surgery has not yet been determined, the potential benefit is supported by data showing a correlation between the use of COX-2 inhibitors and β-blockers with progression-free survival in patients with breast cancer [160,161]. For instance, it was demonstrated in these patients that the peri- and postoperative treatment with anti-inflammatory agents attenuates the impact of surgical wounding on the outgrowth of distant tumors [162]. One week of β-blockade with oral propranolol prior to surgery in patients with breast cancer reduced expression of genes associated with the metastatic potential of tumor cells [163]. A benefit to perioperative β-blockade during surgery-induced stress with respect to breast cancer recurrence and metastases has been also suggested [164]. Similarly, a significant reduction in liver metastases of colon cancer in the context of surgery via β-adrenoceptor blockade and COX-2 inhibition was confirmed in different animal models [165]. CpG-C, a TLR-9 agonist, also markedly improved resistance to colon cancer-associated hepatic metastases in postoperative mice [159]. Other pre-clinical findings have revealed that phosphodiesterase-5 inhibitors could reduce postoperative metastasis by blocking the capacity of surgery-induced MDSC to suppress NK cell cytotoxicity [166]. Although there are not many studies that have tested the impact of targeting the neuro-immune axis in cancer patients before or after surgical procedures, a novel therapeutic paradigm was proposed on the basis of the evidence presented [111]. It declares that the postoperative period represents a window of opportunity during which the patient may be additionally protected against the oncological effects of tumor removal. Upcoming progress in this field should provide effective solutions that would make required unavoidable surgical interventions safer and more useful for the patients.

## 6. Conclusions

Clinical and experimental animal data that show that surgical interventions may boost the emergence of local and distant cancer recurrence have been collected for several decades. However, the notion that surgery-induced wound healing and tissue repair pathways may constitute a risk factor for re-activation of pre-existing dormant malignant cells and the acceleration of growth of distant metastases is not yet widely accepted. Despite this, ongoing investigation of the mechanism of surgery-associated metastatic disease continues revealing new pathways involved in this clinically important and not fully understood phenomenon. In addition to the proven role of surgical stress-induced neurotransmitters and neuromediators, neuroendocrine and stress hormones, inflammatory and tissue remodeling cytokines and chemokines, and angiogenic growth factors, the involvement of new pathways, like Wallerian degeneration, has also been considered. Understanding the biological significance and primary mechanisms of neuroimmune regulation of surgery-accompanying metastasis should advance the field of neuro-immuno-oncology and create a strong foundation for developing novel mechanism-based perioperative prophylactic interventions that protect cancer patients against the adverse effects of surgery and biopsy procedures.

## Figures and Tables

**Figure 1 cells-10-00454-f001:**
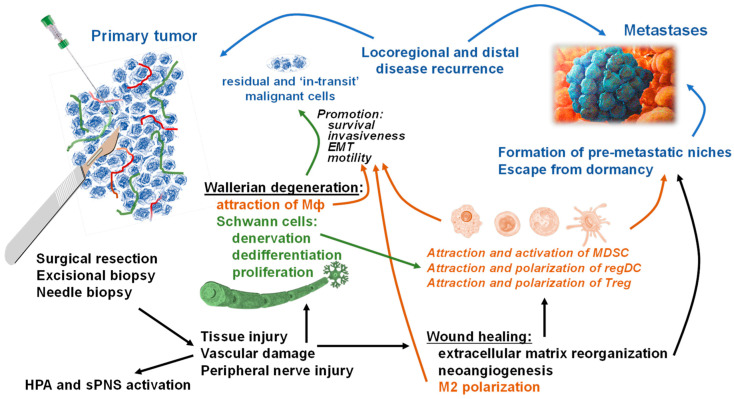
Pathways and mechanisms of accelerated locoregional and distant tumor recurrence after surgical interventions on primary tumors. Resection and biopsy cause unavoidable tissue distraction, damage of blood and lymph vasculature, and injury and trauma of the peripheral neurons. Central systemic effects of surgical stress, together with its psychological components, result in the hypothalamic-pituitary-adrenal (HPA) axis activation and stimulation of the sympathetic branch of the peripheral nervous system (sPNS). Locally, tissue injury and regional hemorrhage, followed by different levels of hypoxemic cellular stress, inflammation, and ischemia, initiate a cascade of tissue repair pathways including wound healing and Wallerian degeneration. The wound healing, after the initial hemostasis and inflammation phases, incorporates extracellular matrix reorganization for remaking new tissue, neoangiogenesis/lymphogenesis for a new network of blood/lymph vessels, and attraction and polarization of regulatory immune cells, such as macrophages (alternatively activated type 2 or M2), for resolving inflammation and augmentation of tissue and vasculature restoration. These pathways phenotypically and functionally resemble the tumor microenvironment characteristics and thus may promote reactivation of dormant malignant cells, the formation of premetastatic niches and the survival and motility of residual and “in-transit” cancerous cells. Wallerian degeneration, prompted by the axonal injury, involves Schwann cell activation–denervation, de-differentiation, and proliferation (‘repair’ phenotype), which is required for the attraction of macrophages (Mф), cleaning the myelin and dead neuronal debris, and axonal regeneration. Resent data revealed that the repair Schwann cells functionally resemble the tumor-activated Schwann cells that can attract and activate myeloid-derived suppressor cells (MDSC), attract conventional dendritic cells (DC) and polarize them into regulatory immunosuppressive DC (regDC), and attract and polarize T cells into the regulatory phenotype (Treg). This pathway also supports local and systemic tumor-associated environments, which favor the establishment and growth of local and distant micrometastases. In addition, activated Schwann cells have been reported to induce the epithelial-mesenchymal transition (EMT) of malignant cells suggesting that tumor pre-activated, as well as axon injury-activated, Schwann cells may boost motility and invasiveness of residual and “in-transit” cancerous cells supporting the formation of the locoregional and distant metastases after surgical excision of primary tumors. (Cancerous cells and their migratory pathways are shown in blue font and blue arrows; immune cells and their effects on malignant cells are shown in brown font and brown arrows; neuroglial Schwann cells and their effects on immune and malignant cells are shown in green font and green arrows. Common pathways connected with primary tumor invasive procedures and tissue injury and repair, and their influences on malignant, neuroglial, and immune cells are shown in black font and black arrows.).

**Table 1 cells-10-00454-t001:** Observations of accelerated postoperative metastatic tumor growth.

Cancer	Species	Notes	Year	Reference
different	human	surgery for primary tumors	1907	[25]
carcinoma	mouse	removal of s.c. tumors	1913	[26]
carcinoma	mouse	localization of tumors at points of injury	1914	[27]
melanoma, sarcoma	mouse	amputation of the tumor-bearing leg	1958	[28]
sarcoma	mouse	pulmonary metastases after amputation of the limb bearing the tumor	1958	[29]
carcinoma of the gastric stump	human	appearance of a second primary lesion after resection for malignancy	1973	[30]
lung carcinoma	mouse	noncurative excision	1976	[3]
testicular cancer	human	cytoreductive surgery	1980	[31]
ovarian cancer	human	cytoreductive surgery	1980	[32]
colon cancer	human	abdominal wall recurrence after colectomy	1983	[33]
melanoma, sarcoma	mouse	tumor cell growth at the surgical wound site	1989	[34]
ovarian cancer	human	cytoreductive surgery	1989	[35]
dermal fibrosarcoma	mouse	spontaneous tumors at abnormal wound repair sites	1990	[36]
melanoma	mice	local tumor removal	1992	[37]
colon cancer	human	recurrence after colectomy	1994	[38]
mammary adenocarcinoma	mouse	tumor injection after surgery, frequency of tumor formation at the site of bone wound	1994	[39]
gastric cancer	human	abdominal wall metastasis after laparoscopic gastroenterostomy	1995	[40]
NSCLC	human	potentially curative resection	1996	[41]
melanoma	mouse	tumor growth in adjacent subcutaneous tissue next to a surgical wound	1998	[42]
breast cancer	human	radical mastectomy	1999	[43]
mammary carcinoma	mouse	lung metastases after open surgery	1999	[44]
lung cancer	human	surgery	2000	[45]
breast cancer	human	mastectomy	2001	[11]
melanoma	mouse	surgery and immunostimulation	2001	[46]
lung carcinoma	mouse	surgical tumor resection	2001	[47]
breast cancer	human	local recurrence in the core needle biopsy site	2001	[18]
colon carcinoma	mouse	lung metastasis after 5 kinds of surgical stress of different degree	2003	[48]
breast cancer	human	needle core biopsy and the incidence of sentinel node metastases	2004	[19]
colorectal cancer	human	local recurrences after curative surgery	2005	[14]
ovarian carcinoma	mouse	laparotomy and mastectomy as surgical stress	2009	[49]
colorectal cancer	human	resection of the primary tumor	2009	[10]
bladder carcinoma	human	surgical removal and rapid metastatic progression	2009	[15]
mammary adenocarcinoma	mouse	resection of primary tumor	2010	[50]
breast cancer	human	surgery	2010	[9]
NSCLC	human	surgery for early-stage cancer	2013	[16,17]
cutaneous melanoma	human	excisional biopsy and local recurrences	2014	[51]
mammary adenocarcinoma	mouse	distant metastases after core needle biopsy	2014	[51]
melanoma	human	sentinel lymph node biopsy and disease recurrence (distant metastases)	2016	[13]
colorectal cancer	human	surgery, partial hepatectomy	2017	[20]
pancreatic ductal adenocarcinoma	human	resection of primary tumor	2018	[21]
colorectal cancer	human	colorectal liver metastases after colorectal surgery	2018	[52]
gastric cancer	human	circulating tumor cells after resection	2018	[53]
hepatocellular carcinoma	human	circulating tumor cells after radical surgery	2018	[23]
lung cancer	human	circulating tumor cells after resection of a lung lobe	2019	[8]
gastric cancer	human	liver metastases after gastrectomy	2020	[54]
pancreatic acinar cell carcinoma	human	gastric and lymph node metastases after pancreatoduodenectomy	2020	[55]
squamous cell carcinoma of tongue	mouse	surgery induces hypoxia, CD11b+ cell infiltration and lymph node metastasis	2020	[56]
colon cancer	mouse	Immunotherapy decreases surgery-induced liver metastases	2020	[57]
papillary thyroid cancer	human	recurrence in subcutaneous area and lymph node metastasis after core needle biopsy	2021	[58]

## Data Availability

Non applicable.

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
