# Peer review of "Neuroimmune Regulation of Surgery-Associated Metastases"

_cells, 2021, doi:10.3390/cells10020454_

Round 1

Reviewer 1 Report

Dear authors,

Thank you for your manuscript titled "Neuroimmune regulation of surgery-associated metastases". In this work you provide a comprehensive overview of the role of the nervous system in augmenting tumour dissemination, progression, and importantly, micrometastasis progression. The concept of perioperative stress upon cancer progression has come and gone and remains riddled with unidentified mechanisms and difficulties, and it is works like this which sum up much of the literature that highlight its dire clinical significance. 

Importantly, you bring attention to previously unreported involvement of Schwann cells, and their reprogrammed "repair" phenotypes, in the promotion of post-surgical metastases...a very interesting and honestly a very reasonable and expected finding (given the widespread nature of these cells and their biological roles). It is surprising this finding has not been previously reported. 

 In sum, this manuscript is very well-written and clearly organized. The data and the references are pertinent and up-to-date.

A few comments:

  1. Given the large amount of information in this review, it would be very useful to include 1-2 figures summarizing this information. Namely, the cascade of events triggering neuro-induced tumour metastasis (+/- surgical stress/wound healing). Specifically, include the role of Schwann cells in this schematic, as you have emphasized these cells.
  2. Conclusion: Have you given any thought to the utility of more novel immunotherapies and their perioperative utility as adjuvant/neoadjuvant additions in counteracting postoperative metastases? Specifically I refer to those that have at least entered human clinical trials...oncolytic viruses (would benefit from the overall global immunosuppressive post-surgical phenotype), ICIs (CTLA4/PD-L1/PD-1), TLR agonists, cell-based therapies and adoptive transfers (NK, CAR-NK, CAR-T, etc). How would some of these approaches fare in the context of this review's topic? 

   Best,

Author Response

Reviewer 1

“In sum, this manuscript is very well-written and clearly organized. The data and the references are pertinent and up-to-date.”

We thank Reviewer 1 for the time and effort spent for evaluating our manuscript.

“Given the large amount of information in this review, it would be very useful to include 1-2 figures summarizing this information. Namely, the cascade of events triggering neuro-induced tumour metastasis (+/- surgical stress/wound healing). Specifically, include the role of Schwann cells in this schematic, as you have emphasized these cells.”

We agree and summarizing Figure 1 with extended figure legend was included in the revised version of the manuscript.

“Conclusion: Have you given any thought to the utility of more novel immunotherapies and their perioperative utility as adjuvant/neoadjuvant additions in counteracting postoperative metastases? Specifically I refer to those that have at least entered human clinical trials...oncolytic viruses (would benefit from the overall global immunosuppressive post-surgical phenotype), ICIs (CTLA4/PD-L1/PD-1), TLR agonists, cell-based therapies and adoptive transfers (NK, CAR-NK, CAR-T, etc). How would some of these approaches fare in the context of this review's topic?”

We thank Reviewer 1 for the important question. We now have Future Directions section which focuses on the role of immunotherapeutic approaches in the field of surgery-associated metastases. We also expanded relevant sections in the test and included references which demonstrated, for instance, that activation of NK cells or depletion of macrophages significantly attenuate formation of surgery-associated metastases in experimental animal models. All human trial approaches mentioned in the reviewer’s notice are reasonable and legible, but it is difficult to imagine that they can be organized in nearest future. We believe that the problem is not enough recognized and accepted by surgeons to talk about clinical trials. Consequently, the goal of our review is to spread more info about the clinical significance of the problem and its potential resolution.   

Reviewer 2 Report

This is a Review article on the impact of the nervous system or neural cells on cancer, specifically on metastases. Please see the following critiques and thoughts for improvement:

  1. The English is of excellent quality and the manuscript readable.
  2. The PDF of the manuscript is 19 pages with only 5 1/2 of these dedicated to text. 12 of the 19 pages are references. The INTRODUCTION and CONCLUSION sections take up nearly 2 of the 5 1/2 pages of actual text leaving only just over 3 pages of "meat" in this manuscript. This imbalance seemed odd and the Review seemed superficial and cursory. Maybe the actual data supporting this Review are so limited that the Review has no "meat" to use.
  3. The Authors provided no figures or tables. Merely reading text with no sort of effort put into illustrating the concepts or providing any data was not very inspiring.
  4. Much of the language in this Review was passive and uncertain. Words such as "may be", "it is possible", "could provide", and "we can speculate" are seem numerous times in the Review. These phrases imply that the Review is not a synthesis of facts or knowledge but more so a collection of theories and suggestions that have not been proven or even studied.
  5. Per comment #4 above, the Authors make statements about the concept of neuroimmune regulation of metastases and describe several aspects as "never been suggested" or "never been considered". With this spin on the manuscript it reads more like a manuscript telling the Reader what we DO NOT know rather than what we DO know.
  6. In the Introduction, the Authors begin with a statement that 1.8 million (USA) and 20 million (worldwide) cases of cancer are diagnosed annually and that a majority are associated with metastases. This does not seem to be a true statement. Are the Authors suggesting a majority of cancers present in advanced stages with metastases? Is this accurate? The reference for this statement does state that 1.8 million are diagnosed with cancer in the USA, but it also suggests that number is 17 million worldwide (rather than 20 million). The implication that a majority have metastases is not seen in this reference and is likely inaccurate.
  7. The CONCLUSION section is too long and does not tie up the Review. It is merely an extra DISCUSSION paragraph.
  8. Thank you for the opportunity to assess this manuscript.

Author Response

Reviewer 2

“The PDF of the manuscript is 19 pages with only 5 1/2 of these dedicated to text. 12 of the 19 pages are references. The INTRODUCTION and CONCLUSION sections take up nearly 2 of the 5 1/2 pages of actual text leaving only just over 3 pages of "meat" in this manuscript. This imbalance seemed odd and the Review seemed superficial and cursory. Maybe the actual data supporting this Review are so limited that the Review has no "meat" to use.”

We appreciate Reviewer 2 comprehensive analysis of the manuscript. The reviewer is correct that actual data demonstrating neuro-immune pathway in the regulation of surgery-induced metastases are very limited. This is what makes this review unique, timely and interesting. With this review we want to provide new, unpublished but important information to stimulate analytical thinking and discussion, and, possibly, more research in the new field.

“The Authors provided no figures or tables. Merely reading text with no sort of effort put into illustrating the concepts or providing any data was not very inspiring.”

We appreciate Reviewer 2 suggestion and included a summary figure and a table in the revised version of the manuscript.

“Much of the language in this Review was passive and uncertain. Words such as "may be", "it is possible", "could provide", and "we can speculate" are seem numerous times in the Review. These phrases imply that the Review is not a synthesis of facts or knowledge but more so a collection of theories and suggestions that have not been proven or even studied.”

We again agree with Reviewer 2 that the review is unique in terms of its innovation and freshness. It is correct that the main concept of participation of neuroglial cells in surgery-associated tumor cell spreading is very novel and is based only a few available publications. It definitely requires further confirmation and verification in different models, and the review presents a basis for this kind of further research. The first part of the review provides a basis knowledge of clinically and experimentally verified facts, which form a fundament for a new concept and a new point of view on a novel pathway participating in surgery-induced formation of residual and distant metastases.  

“Per comment #4 above, the Authors make statements about the concept of neuroimmune regulation of metastases and describe several aspects as "never been suggested" or "never been considered". With this spin on the manuscript it reads more like a manuscript telling the Reader what we DO NOT know rather than what we DO know.”

We again agree with Reviewer 2: the review provides several innovative directions for solving the problem and suggests new avenues for future studies. We hope that this feature of the review would stimulate wider discussion and allow selection of the best way to design new experimental and clinical studies. Sorry for using the reviewer’s terminology, but the readers might prefer fresh “meat” rather than several times digested “meat”. We believe that the readers will make their own choice, while our job is to provide several legible and scientifically-sound choices.

“In the Introduction, the Authors begin with a statement that 1.8 million (USA) and 20 million (worldwide) cases of cancer are diagnosed annually and that a majority are associated with metastases. This does not seem to be a true statement. Are the Authors suggesting a majority of cancers present in advanced stages with metastases? Is this accurate? The reference for this statement does state that 1.8 million are diagnosed with cancer in the USA, but it also suggests that number is 17 million worldwide (rather than 20 million). The implication that a majority have metastases is not seen in this reference and is likely inaccurate.”

We agree that the statement that most of cancers are associated with metastases is an overestimation. The text was corrected and now uses the reference stated that fifty percent or more of malignant tumors have metastasized prior to clinical presentation. In regard to cancer statistics, Intl Agency for Research on Cancer and WHO reported 18.1 million new cancer cases in 2018 (https://www.who.int/cancer/PRGlobocanFinal.pdf) and 19 292 789 new cancer cases for 2020 (https://gco.iarc.fr/today/data/factsheets/cancers/39-All-cancers-fact-sheet.pdf)  Since the global cancer burden continues to grow, in our review we cautiously stated “almost 20 million cases worldwide” since, if accepted, it will be published in 2021 and still be read in 2022.

“The CONCLUSION section is too long and does not tie up the Review. It is merely an extra DISCUSSION paragraph.”

We thank Reviewer 2 for this critical point. We agree and re-write Conclusions. It is correct that ‘old’ conclusions look like a discussion of potential therapeutic applications related to the way to minimize tumor spreading after surgical interventions. This part of the review was transformed into Future directions in the revised version of the manuscript.

We want to thank Reviewer 2 for very helpful and significant contribution in improving the manuscript.

Reviewer 3 Report

Metastasis formation remain still the major clinical challenges in cancer treatment. A detailed understanding of the underlying mechanisms is pivotal for developing more effective preventive and therapeutic strategies. The complex metastatic cascade commences with the emigration of tumor cells from the primary tumor and ends with colonization and outgrowth of clinically detectable metastases at secondary sites in a process reminiscent of Epithelial-Mesenchymal-Transition.

The authors present in this review the role of the PNS as a major contributor to cancer development and metastasis, and how may PNS elements be involved in surgery-induced metastasis spreading. From this reviewer's point of view, although this hypothesis demands attention, the authors do not go beyond theoretical reflection motivated by observation. Theoretical models, such as those presented in this review, should make valid predictions under a variety of laboratory and clinical conditions. The mitogenic impact on residual cancer foci caused by surgery is an observation, fundamentally explicable by circulating factors. These approaches will not be sufficient to influence treatment decisions until a mechanistic, research-dependent verification of these testable ideas is made.

Logically, a randomized trial in which cancer patients are assigned to surgery or no surgery is not possible. Authors should propose clinical trial designs that should be performed to validate their hypothesis. Impact reduction data of surgical wounding on the outgrowth of distant tumors using anti-inflammatory medication, β-blockers and phosphodiesterase-5 inhibitors should progress from preclinical and retrospective clinical studies to prospective clinical studies.

Author Response

Reviewer 3

“The authors present in this review the role of the PNS as a major contributor to cancer development and metastasis, and how may PNS elements be involved in surgery-induced metastasis spreading. From this reviewer's point of view, although this hypothesis demands attention, the authors do not go beyond theoretical reflection motivated by observation. Theoretical models, such as those presented in this review, should make valid predictions under a variety of laboratory and clinical conditions. The mitogenic impact on residual cancer foci caused by surgery is an observation, fundamentally explicable by circulating factors. These approaches will not be sufficient to influence treatment decisions until a mechanistic, research-dependent verification of these testable ideas is made.”

Reviewer 3 is absolutely correct: mechanism-based verification of innovative concept is needed. One goal of the review is to justify these studies and provide a few directions for future research. We believe that introduction of a new model that appears from old and recent observations should promote critical thinking, discussion and optimization of future experiments. We realize that this format of the review is somehow unusual but we rely on the fact that readers’ interest are wide-ranging and that the review may attract new researchers and new expertise in the field of neuro-oncology and help solving a clinically important and urgent medical problem.

“Logically, a randomized trial in which cancer patients are assigned to surgery or no surgery is not possible. Authors should propose clinical trial designs that should be performed to validate their hypothesis. Impact reduction data of surgical wounding on the outgrowth of distant tumors using anti-inflammatory medication, β-blockers and phosphodiesterase-5 inhibitors should progress from preclinical and retrospective clinical studies to prospective clinical studies.”

We are 100% with Reviewer 3 on this important notion about well-timed clinical trials which are necessary to move the discussed area of neuro-immune-oncology. The rapid advance in our understanding of cancer metastasis at molecular and cellular levers as well as signaling pathways provides numerous potential targets for the intervention of cancer metastasis. In addition to agents mentioned by Reviewer 3, this is especially true in terms of intervention with inflammatory, neurological and immunological pathways, biochemical processes and signaling pathways involved in cell detachment, migration, invasion, adhesion and cancer cell communication with different stromal elements. However, we are not in a position to design surgical clinical trials at this time. The goal of the review is to stress the problem and introduce the problem to clinical and biomedical researchers who is in a position to appreciate the importance of the problem, make a rational judgement and conduct research to progress from preclinical to retrospective and prospective clinical investigation.

Round 2

Reviewer 2 Report

This is a revised review manuscript discussing surgery-mediated metastases. Please consider the following comments:

  1. The Review continues to suffer from a lack of focus. It is challenging to read as it has a rambling nature to the text. After reading it, this Reviewer is not at all convinced that "the peripheral nervous system has been recognized as a major contributor to cancer development and metastasis" as the Authors state. The PNS causes cancer to develop?? If this is an absolute fact and the PNS is a "major contributor" then please provide these unequivocal data. 
  2. The Authors responded to the previous review in a way that came across as dismissive. This Reviewer feels a Review article should provide facts about the topic the Authors hope to convey, namely, in this case, the "Neuroimmune Regulation of Surgery-Associated Metastases". The Authors use the words "unique" and "fresh" to describe their Review but the Review reads as a theoretical piece rather than an authoritative work (as it should be). Unfortunately, the manuscript does a very poor job of educating the reader. A Review is not meant to be a set of theories to recommend for further study. Again, the fact that the Authors make this statement in their Abstract "very little is known about the contribution of tumor-associated neuronal and neuroglial elements in the metastatic disease related to surgical trauma and wound healing. Specifically, although numerous clinical and experimental data suggest that biopsy and surgery-induced wound healing can promote survival and metastatic spread of residual and dormant malignant cells, the involvement of the tumor-associated neuroglial cells in the formation of metastases following tissue injury has never been considered." suggests that this angle of their Review doesn't have enough data to support it as a Review.
  3. The Authors have added this statement to the Introduction; "Fifty percent or more of malignant tumors in humans have metastasized prior to clinical presentation [3,4]." yet these references (#3 and #4) are 40-45 years old ... from a journal article published in 1976 and from a textbook published in 1983. Surely, something has changed. This number is lower that 50%.
  4. In the new table, the Authors should remain consistent with the "Species" ... Human, Rat, Mouse. Mice is plural so would be incorrect. In that same Table, there is a column for "Procedures" however what is written in that column varies. Some are procedures such as "amputation" or "cytoreductive surgery" but others are not procedures such as "surgery induces hypoxia, CD11b+ cell infiltration and lymph node metastasis" or "increased proliferation of tumor cells locally and in
    distant metastases after surgery". Much like the Species column, this Procedures column should also be consistent and depict actual procedures.
  5. The Figure has many arrows. Are these arrows supported by data? Or are they theories of what "might" be possible?

Thank you for asking for my input.

Author Response

Point-by point respond to Reviewer 2 concerns:

  1. The Review continues to suffer from a lack of focus. It is challenging to read as it has a rambling nature to the text. After reading it, this Reviewer is not at all convinced that "the peripheral nervous system has been recognized as a major contributor to cancer development and metastasis" as the Authors state. The PNS causes cancer to develop?? If this is an absolute fact and the PNS is a "major contributor" then please provide these unequivocal data. 

We agree with Reviewer 2 that our statement that the PNS is “a major” contributor to cancer development and progression is an overemphasis. We have corrected this overstatement throughout the text in the revised version of the manuscript by saying that the PNS is another contributor in tumor development and that related mechanisms are not yet well understood. We want to thank Reviewer 2 for helpful and strong suggestions for improving the manuscript.

  1. The Authors responded to the previous review in a way that came across as dismissive. This Reviewer feels a Review article should provide facts about the topic the Authors hope to convey, namely, in this case, the "Neuroimmune Regulation of Surgery-Associated Metastases". The Authors use the words "unique" and "fresh" to describe their Review but the Review reads as a theoretical piece rather than an authoritative work (as it should be). Unfortunately, the manuscript does a very poor job of educating the reader. A Review is not meant to be a set of theories to recommend for further study. Again, the fact that the Authors make this statement in their Abstract "very little is known about the contribution of tumor-associated neuronal and neuroglial elements in the metastatic disease related to surgical trauma and wound healing. Specifically, although numerous clinical and experimental data suggest that biopsy and surgery-induced wound healing can promote survival and metastatic spread of residual and dormant malignant cells, the involvement of the tumor-associated neuroglial cells in the formation of metastases following tissue injury has never been considered." suggests that this angle of their Review doesn't have enough data to support it as a Review.

The reviewer point is well taken. However, the authors believe that there are different types of review papers with different goals and for different groups of readers. This review is based on published data and our own published results. We agree that all published data do not represent a big pool of information yet, but they still support a new concept discussed in the review. The goal of the review is to examine and consider new ideas in an important field with the hope that this might help accelerating development of procedures to minimize formation of metastases associated with invasive diagnostic and treatment procedures, which can make these surgical interventions safer and more beneficial for the patients. We realize that our view on the significance of reviews in education and research may be different, but we believe that a variety of review types may better serve a variety of reader groups.

  1. The Authors have added this statement to the Introduction; "Fifty percent or more of malignant tumors in humans have metastasized prior to clinical presentation [3,4]." yet these references (#3 and #4) are 40-45 years old ... from a journal article published in 1976 and from a textbook published in 1983. Surely, something has changed. This number is lower that 50%.

This is another important point of concern. Since we fail to find any other published references on this topic, we have corrected the test in the revised version of the manuscript: “It was estimated earlier that almost fifty percent of malignant tumors in humans have metastasized prior to clinical presentation [3,4], although the number may be lower and is certainly cancer dependent.”

  1. In the new table, the Authors should remain consistent with the "Species" ... Human, Rat, Mouse. Mice is plural so would be incorrect. In that same Table, there is a column for "Procedures" however what is written in that column varies. Some are procedures such as "amputation" or "cytoreductive surgery" but others are not procedures such as "surgery induces hypoxia, CD11b+ cell infiltration and lymph node metastasis" or "increased proliferation of tumor cells locally and in distant metastases after surgery". Much like the Species column, this Procedures column should also be consistent and depict actual procedures.

We agree with Reviewer 2 and apologize for our mistake. “mice” were replaced by “mouse”. The term “procedures” is incorrect in this table. We intended to briefly mention what is important to understand the results. It was also corrected.

  1. The Figure has many arrows. Are these arrows supported by data? Or are they theories of what "might" be possible?

We thank Reviewer 2 for this important concern. All pathways presented on the summary figure are based on published papers. Only one mechanism – attraction and polarization of T cells by tumor-activated Schwann cells, is not published yet. The manuscript is in preparation.